**∂ | Open Peer Review** | Mycology | Research Article

# Antifungal susceptibility profile and local epidemiological cut-off values of *Yarrowia* (*Candida*) *lipolytica*: an emergent and rare opportunistic yeast

Jinhan Yu,[1,2,3] Xueqing Liu,[4] Dawen Guo,[5] Wenhang Yang,[1,2] Xinfei Chen,[1,2,3] Guiling Zou,[6] Tong Wang,[1,2] Shichao Pang,[7] Ge Zhang,[1,2] Jingjing Dong,[1,2] Yingchun Xu,[1,2] Ying Zhao,[1,2] on behalf of the National China Hospital Invasive Fungal Surveillance Network (CHIF-NET)

**ABSTRACT**  The antifungal susceptibility profile and epidemiological cut-off values (ECOFFs) of *Yarrowia lipolytica*, a rare opportunistic yeast, remain unclear. We conducted a comprehensive multi-method study on clinical isolates from various central hospitals, based on the China Hospital Invasive Fungal Surveillance Network (2009–2022). Our objective was to evaluate the antifungal susceptibility of *Y. lipolytica*, establish its local ECOFFs (L-ECOFFs), and compare the performance of the ATB FUNGUS 3 (ATB), Sensititre YeastOne (SYO), and minimum inhibitory concentration (MIC) test strip (MTS) with that of the broth microdilution (BMD) method. L-ECOFFs were established using ECOFFinder, and we examined *ERG11* mutations to assess the reliability of the L-ECOFFs. The L-ECOFF for fluconazole was 8 µg/mL. Non-wild-type isolates of antifungal drugs, such as flucytosine and azoles, were exclusively isolated from patients. Additionally, we detected that four strains with the *ERG11* A395T mutation (azole MIC >L-ECOFF) may be associated with the exposure to azole drugs. For azoles, ATB showed the highest essential agreement with the BMD (98.18%–100%), followed by SYO (85.45%–100%). However, ATB could not detect susceptibility to echinocandins, while SYO exhibited the highest agreement (98.18%–100%) in detecting echinocandin susceptibility. Our findings indicate that acquired azole cross-resistance has emerged despite *Y. lipolytica* infections being rare. This research provides crucial antifungal susceptibility data and establishes the initial L-ECOFFs for *Y. lipolytica*. The SYO is recommended as the optimal laboratory antifungal susceptibility testing method for *Y. lipolytica*, followed by ATB, whereas the use of MTS requires caution. We hope that this study will facilitate improved clinical management of *Y. lipolytica* infections.

**IMPORTANCE**  *Yarrowia lipolytica*, also known as *Candida lipolytica*, is an emerging opportunistic "rare pathogenic yeast". Due to the limited data on its antifungal susceptibility, the clinical treatments become challenging. Based on the China Hospital Invasive Fungal Surveillance Network (2009–2022), we conducted a comprehensive multi-method study on clinical isolates from various central hospitals. This study is currently the largest study carried out to assess the antifungal susceptibility of *Y. lipolytica*. It is also the first to establish local epidemiological cut-off values (L-ECOFFs), identify its *ERG11* mutations, and assess the consistency between the three prevalent commercial antifungal susceptibility testing methods and the broth microdilution method. We recommend the Sensititre YeastOne as the best option for antifungal susceptibility testing for *Y. lipolytica*, followed by the ATB FUNGUS 3. Nevertheless, practitioners should use the MIC test strip with discretion.

Address correspondence to Ying Zhao, zhaoying28062806@163.com, or Yingchun Xu, xycpumch@139.com.

The authors declare no conflict of interest.

See the funding table on p. 11.

**KEYWORDS** *Yarrowia lipolytica*, *Candida lipolytica*, antifungal susceptibility testing, minimum inhibitory concentration, epidemiological cut-off value, *ERG11*, azole cross-resistance

*Yarrowia lipolytica* [(Wick., Kurtzman & Herman) Van der Walt & Arx, 1980] is one of the most representative dimorphic unconventional yeasts (1). Currently, it is clinically and conventionally referred to as *Candida lipolytica* (2). *Y. lipolytica* is a new emerging opportunistic "rare yeast" that (i) occurs in low prevalence (<1% of clinical *Candida* infections) and (ii) expresses an elevated minimum inhibitory concentration (MIC) for at least one class of antifungal drugs (3). Since three cases of eye infection caused by *Y. lipolytica* were reported in France in 1976, more than 30 relevant pathogenic studies and case reports have been published (4–12). Recently, *Y. lipolytica* has been reported to cause nosocomial outbreaks, foodborne infections, breakthrough fungemia, and catheter-related infections that may be complicated by septic shock in immunocompromised patients and even those co-infected with severe acute respiratory syndrome coronavirus 2 (4, 9–11, 13).

At present, there are few studies and limited data regarding *Y. lipolytica* infections. The epidemiological cut-off values (ECVs/ECOFFs) and clinical breakpoints for the international standard methods of the European Committee for Antimicrobial Susceptibility Testing (EUCAST) and Clinical and Laboratory Standard Institute (CLSI) have not yet been established for *Y. lipolytica* (3). The treatment of infections caused by *Y. lipolytica* is uncertain owing to limited clinical experience and lack of susceptibility data, leading to a high frequency of therapeutic failure (14).

Azole drugs (fluconazole, itraconazole, voriconazole, and posaconazole) are effective against a broad range of fungal species, making them a common treatment option for many types of fungal infections. It is also important for the treatment of invasive *Y. lipolytica* infections (4, 6, 15). The fungal cytochrome P450 lanosterol 14α-demethylase (encoded by the *ERG11* gene) is required for the biosynthesis of fungal-specific ergosterol and is the target of azole drugs (16). Notably, the *Y. lipolytica ERG11* mutation (A395T) has been identified exclusively in the fluconazole-resistant strain CBS 18115 (17). Its presence in other isolates remains to be determined.

Several techniques for antifungal susceptibility testing are available, such as the broth microdilution (BMD) reference method recommended by the CLSI and EUCAST, commercial techniques that use colorimetric endpoints (e.g., Sensititre YeastOne [SYO] assay), agar-based methods that use concentration gradients of antifungals that diffuse into the growth media (e.g., E-test and Liofilchem MIC test strips [MTS]), and ATB FUNGUS 3 (ATB) test strips. Commercial methods are standardized, effective, simple, and easy to implement compared to gold standard reference techniques, which are tedious, time-consuming, and limited to a few centers (18). Therefore, it is essential to evaluate these commercial methods and determine their ability in providing MICs that agree with those of reference methods.

The aims of this study were to (i) generate data on the antifungal susceptibility profile of *Y. lipolytica* in China over the past 10 years; (ii) establish local ECOFFs (L-ECOFFs); (iii) identify *Y. lipolytica ERG11* mutations; and (iii) compare the MICs obtained using the CLSI BMD method with those generated using the SYO, MTS, and ATB methods.

## RESULTS

The antifungal susceptibility testing of three isolates including CGMCC 2.3222 isolated from flies, CGMCC 2.1556 isolated from garbage stations in East China, and CGMCC 2.1502 isolated from the Faculty of Agriculture, Kyoto University, Japan, was tested at 30°C, as these isolates exhibited poor growth at 35°C (Table S1). The MICs of the nine antifungal drugs against the 74 tested *Y. lipolytica* isolates at 35°C are presented in Table 1. Nonlinear regression fitting was performed on the MIC distribution data measured using the BMD method for the nine antifungal drugs, and the fitting line presented a

**TABLE 1** Distribution of the minimum inhibitory concentration values for clinical and non-clinical isolates of *Yarrowia lipolytica* and the local epidemiological cut-off values for the broth microdilution and commercial methods (μg/mL)[a]

| Antifungal agent | Method | L-ECOFFs | Non-WT (%) | 0.002 | 0.004 | 0.008 | 0.016 | 0.03 | 0.06 | 0.12 | 0.25 | 0.5 | 1 | 2 | 4 | 8 | 16 | 32 | 64 | 128 | 256 | >256 |
|---|---|---|---|---|---|---|---|---|---|---|---|---|---|---|---|---|---|---|---|---|---|---|
| Anidulafungin | BMD | 1 | 0 | | | | | | 6 | 19 | **31** | 18 | | | | | | | | | | |
| | SYO | 0.5 | 1.4 | | | | | 3 | 8 | **50** | 9 | 3 | 1 | | | | | | | | | |
| | MTS | 1 | 1.4 | | | | | 1 | 3 | 7 | **23** | 22 | 17 | 1 | | | | | | | | |
| Micafungin | BMD | 2 | 0 | | | | | 1 | 0 | 4 | 26 | **39** | 4 | | | | | | | | | |
| | SYO | 2 | 0 | | | | | | 1 | 7 | 17 | **46** | 3 | | | | | | | | | |
| | MTS | 1 | 0 | 1 | 1 | 0 | 0 | 0 | 7 | 24 | **30** | 11 | 3 | | | | | | | | | |
| Caspofungin | BMD | 1 | 0 | | | 0 | 0 | 1 | 1 | 21 | **38** | 13 | | | | | | | | | | |
| | SYO | 1 | 0 | | | | | | 3 | 16 | **34** | 19 | 2 | | | | | | | | | |
| | MTS | 8 | 0 | | | | | | | 1 | 0 | 7 | 24 | **36** | 5 | 1 | | | | | | |
| Flucytosine | BMD | 8 | 14.9 | | | | | | | | 4 | 3 | 4 | **19** | 18 | 15 | 6 | 1 | 4 | | | |
| | SYO | 128 | 0 | | | | | | | | 2 | 1 | 4 | 6 | 11 | 11 | 6 | **18** | 7 | 8 | | |
| | MTS | 256 | 0 | 1 | 0 | 0 | 0 | 0 | 0 | 1 | 0 | 0 | 1 | 6 | 7 | 14 | 9 | 2 | **33** | 8 | | |
| | ATB | 64 | 0 | | | | | | | | | | | | 8 | 0 | **35** | 31 | | | | |
| Posaconazole | BMD | 4 | 16.2 | | | | | | 5 | 5 | 13 | 16 | **20** | 8 | 0 | 11 | 1 | | | | | |
| | SYO | 2 | 17.6 | | | | 1 | 0 | 0 | 3 | 16 | **29** | 11 | 1 | 1 | 0 | 11 | 12 | | | | |
| | MTS | 2 | 25.7 | | | | | 1 | 0 | 2 | 6 | **19** | 16 | 11 | 7 | 3 | 4 | 1 | 4 | | | |
| Voriconazole | BMD | 0.5 | 17.6 | | | | | 10 | 14 | **22** | 6 | 9 | 1 | 2 | 10 | | | | | | | |
| | SYO | 0.25 | 20.3 | | | | 2 | 15 | **19** | 15 | 8 | 1 | 2 | 6 | 4 | 2 | | | | | | |
| | MTS | 0.25 | 20.3 | 1 | 2 | 1 | 3 | 8 | **23** | 15 | 6 | 1 | 2 | 0 | 5 | 5 | 1 | 0 | 1 | | | |
| | ATB | 0.5 | 17.6 | | | 1 | | | 21 | **25** | 11 | 4 | 1 | 2 | 9 | 1 | | | | | | |
| Itraconazole | BMD | 1 | 16.2 | | | | | | 1 | 7 | **34** | 17 | 3 | 0 | 6 | 5 | 1 | | | | | |
| | SYO | 1 | 16.2 | | | | | 2 | 1 | 19 | **26** | 12 | 2 | 0 | 1 | 0 | 0 | 11 | | | | |
| | MTS | 2 | 20.3 | | | | 1 | 0 | 1 | 2 | 13 | **19** | 15 | 8 | 5 | 0 | 2 | 0 | 8 | | | |
| | ATB | 1 | 18.9 | | | | 1 | | | 23 | 25 | 5 | 7 | 0 | 1 | 13 | | | | | | |
| Fluconazole | BMD | 8 | 18.9 | | | | | | | | | 2 | 9 | **22** | 17 | 10 | 1 | 6 | 7 | | | |
| | SYO | 8 | 28.4 | | | | | | | | | 1 | 1 | **25** | 20 | 6 | 6 | 2 | 7 | 6 | | |
| | MTS | 8 | 28.4 | | | | | | | | | 1 | 5 | 11 | **27** | 9 | 7 | 2 | 0 | 0 | 0 | 12 |
| | ATB | 16 | 18.9 | | | | | | | | | 1 | 5 | 20 | **27** | 8 | 0 | 5 | 9 | | | |
| Amphotericin B | BMD | 4 | 0 | | | | | | | | | 6 | **63** | 5 | | | | | | | | |
| | SYO | 4 | 0 | | | | | | | | 2 | 14 | **43** | 15 | | | | | | | | |
| | MTS | 2 | 0 | 1 | 0 | 0 | 0 | 0 | 6 | 15 | 24 | **25** | 3 | | | | | | | | | |
| | ATB | 2 | 0 | | | | | | | | 1 | **65** | 8 | | | | | | | | | |

[a]BMD: broth microdilution; SYO: Sensititre YeastOne; ATB: ATB FUNGUS 3; MTS: Liofilchem minimum inhibitory concentration (MIC) test strip; L-ECOFFs: local epidemiological cut-off values; non-WT: non-wild-type. Off-scale MICs were converted to the next concentration, up or down, and included in the analysis. L-ECOFFs were determined using the derivatization method and ECOFFinder software with a 97.5% confidence interval. The highest number in each row (showing the most frequent, or mode, MIC) is in bold.

lognormal distribution (Fig. 1). Additionally, the L-ECOFFs that were calculated at a 97.5% confidence interval are also listed in Table 1.

The proportion of non-WT isolates of the four azoles were 12/74 (16.22%) for posaconazole, 13/74 (17.57%) for voriconazole, 12/74 (16.22%) for itraconazole, and 14/74 (18.92%) for fluconazole. These non-WT isolates were all isolated from clinical samples. Furthermore, the proportion of non-WT isolates for flucytosine was 14.86% (11/74), and all tested isolates were found to be WT for echinocandin drugs.

Mutations of the *YALI0_B05126g* were screened, and six non-synonymous and one synonymous mutation were found (Table 2). Analyses of the *YALI0_B05126g* sequence among 135 isolates demonstrated that only four strains (Y04, Y08, Y26, and Y27), which were isolated from peripheral blood or venous catheter, exhibited the A395T (Tyr132Phe) mutation. These four isolates displayed MIC values surpassing the L-ECOFFs for all four azoles tested. The patient infected with the Y04 isolate had been treated with itraconazole for a period of 11 days. Moreover, the patient from whom the Y08 was isolated, as well as the patient from whom the Y26 and Y27 were isolated, had a history of receiving fluconazole via intravenous administration. Additionally, the Y08, Y26, and Y27

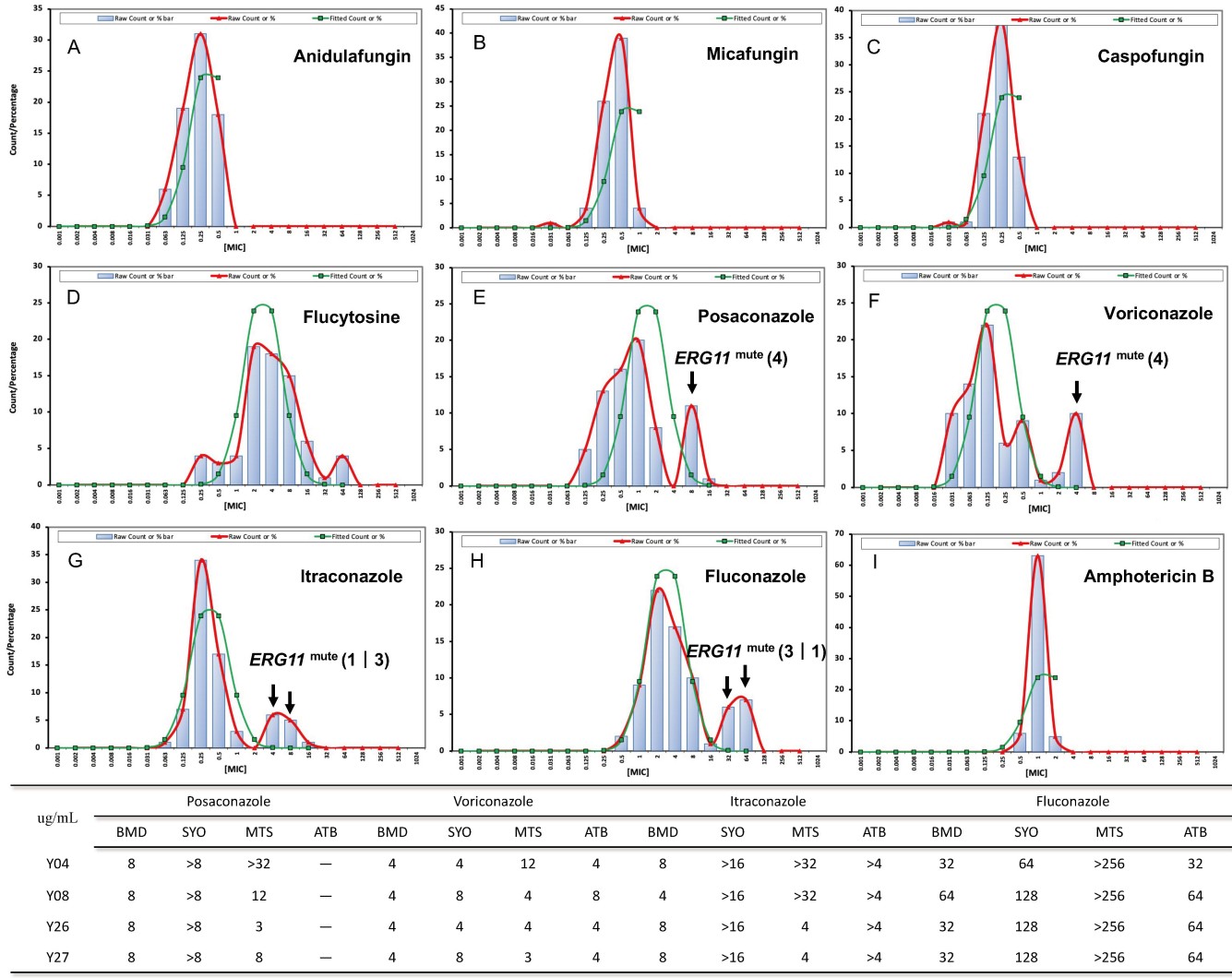

| ug/mL | Posaconazole | | | | Voriconazole | | | | Itraconazole | | | | Fluconazole | | | |
|---|---|---|---|---|---|---|---|---|---|---|---|---|---|---|---|---|
| | BMD | SYO | MTS | ATB | BMD | SYO | MTS | ATB | BMD | SYO | MTS | ATB | BMD | SYO | MTS | ATB |
| Y04 | 8 | >8 | >32 | — | 4 | 4 | 12 | 4 | 8 | >16 | >32 | >4 | 32 | 64 | >256 | 32 |
| Y08 | 8 | >8 | 12 | — | 4 | 8 | 4 | 8 | 4 | >16 | >32 | >4 | 64 | 128 | >256 | 64 |
| Y26 | 8 | >8 | 3 | — | 4 | 4 | 4 | 4 | 8 | >16 | 4 | >4 | 32 | 128 | >256 | 64 |
| Y27 | 8 | >8 | 8 | — | 4 | 8 | 3 | 4 | 8 | >16 | 4 | >4 | 32 | 128 | >256 | 64 |

FIG 1 Nonlinear regression fitting was performed on the minimum inhibitory concentration distribution data for nine antifungal drugs measured by the broth microdilution method. (A) anidulafungin; (B) micafungin; (C) caspofungin; (D) flucytosine; (E) posaconazole; (F) voriconazole; (G) itraconazole; (H) fluconazole; (I) amphotericin B. *ERG11* mutant strains with *ERG11* A395T mutation. The black arrow points to the minimum inhibitory concentration of the mutant strain, and the number of strains is represented in brackets.

**TABLE 2** *YALI0_B05126g* (*ERG11*) gene mutations among 135 *Yarrowia lipolytica* isolates (including all clinical isolates from the NCBI)[a]

| Mutation type | Nucleotide mutation | Amino acid mutation | CGMCC 2.1556 | CGMCC 2.1711 | CGMCC 2.1713 | CICC 1664 | NCYC 3727 | Y04 | Y06 | Y08 | Y26 | Y27 | Y47 |
|---|---|---|---|---|---|---|---|---|---|---|---|---|---|
| Missense | 395A > T | Tyr132Phe | 0;0 | 0;0 | 0;0 | 0;0 | 0;0 | **1;1** | 0;0 | **1;1** | **1;1** | **1;1** | 0;0 |
| Missense | 409T > A | Ser137Thr | 0;0 | 0;0 | 0;0 | 0;0 | **1;1** | 0;0 | 0;0 | 0;0 | 0;0 | 0;0 | 0;0 |
| Synonymous | 807C > T | Asp269Asp | **0;1** | 0;0 | 0;0 | 0;0 | 0;0 | 0;0 | 0;0 | 0;0 | 0;0 | 0;0 | 0;0 |
| Missense | 874G > T | Ala292Ser | 0;0 | **1;1** | **1;1** | 0;0 | 0;0 | 0;0 | 0;0 | 0;0 | 0;0 | 0;0 | 0;0 |
| Missense | 975T > A | Asp325Glu | 0;0 | 0;0 | 0;0 | **1;1** | 0;0 | 0;0 | 0;0 | 0;0 | 0;0 | 0;0 | 0;0 |
| Missense | 1303G > T | Ala435Ser | 0;0 | 0;0 | 0;0 | 0;0 | 0;0 | 0;0 | 0;0 | **0;1** | **0;1** | **0;1** | 0;0 |
| Missense | 1401T > G | Ile467Met | 0;0 | 0;0 | 0;0 | 0;0 | 0;0 | 0;0 | **1;1** | 0;0 | 0;0 | 0;0 | **1;1** |

[a]Note: 0;0: wild type; 0;1: heterozygous mutation; 1;1: homozygous mutation. the Y04 patient was treated with 350 mg itraconazole intravenously for 11 days; no antifungal drugs were used for the Y06 patient; two patients involved in Y08, Y26, and Y27 had a history of receiving fluconazole via intravenous administration.

also harbored a G1303T (Ala435Ser) mutation. The antifungal susceptibility of the four strains with the *ERG11* A395T mutation is shown in Fig. 1.

The MIC value necessary to inhibit at least 90% of the clinical isolates (MIC$_{90}$) for echinocandins was 0.5 µg/mL, and that for amphotericin B remained at 1 µg/mL. The MIC for flucytosine showed considerable variation, ranging from 0.25 µg/mL to 64 µg/mL (Table 3). For the detection of azole susceptibility, the ATB showed the highest essential agreement (EA) with the BMD (98.18%–100%), followed by the SYO (85.45%–100%). The EA of the MTS for detecting voriconazole was 90.90%, and for the other two azole drugs, the EA was ≤80%. The ATB cannot detect susceptibility to echinocandins, whereas the SYO had the highest detection EA (98.18%–100%) for echinocandins. The EA of the MTS in detecting the antifungal susceptibility to echinocandins was 47.27%–98.18% (Table 3).

We also tested the antifungal susceptibility of isolates from non-clinical sources (Table S2). The EA of the SYO for the detection of echinocandin was 100%. For the detection of azole drugs and amphotericin B, the EA of the ATB method could reach 100%, and the EA of detection by the SYO was 89.47%–100%. The EA of the three commercial methods for flucytosine was <80%.

## DISCUSSION

Emerging pathogens and environmental fungi are a class of pathogens that pose a major threat to public health. These pathogens could colonize the surface of the hospital environment and medical equipment and spread infection along with diagnosis and treatment activities. The environmental fungus *Y. lipolytica* is an important emerging rare pathogen. Based on the CHIF-NET program (2009–2022), we conducted a multi-method study on the antifungal susceptibility of *Y. lipolytica* and established its L-ECOFFs. This study fills the gap in comprehensive antifungal susceptibility profiles of *Y. lipolytica* in China, with the aim of providing a valuable reference for clinical therapeutics.

The European Society of Clinical Microbiology and Infectious Diseases Fungal Infection Study Group and the European Confederation of Medical Mycology pointed out that the MIC of fluconazole for *Y. lipolytica* is higher than that for *Candida albicans* (19). The Westerdijk Fungal Biodiversity Institute (formerly Centraal Bureau voor Schimmelcultures, CBS-KNAW) and Institute Pasteur conducted EUCAST BMD and found high MICs for flucytosine, wherein the majority (14/34) was susceptible to fluconazole, and all to voriconazole, posaconazole, caspofungin, and amphotericin B in *Y. lipolytica* (9). In contrast, a separate study on 27 bloodstream infection isolates indicated that *Y. lipolytica* had high MICs for fluconazole, itraconazole, and posaconazole (20). The majority were susceptible toward amphotericin B and voriconazole (98.9%), whereas anidulafungin and micafungin exhibited resistance in 74.1% (20). The National Reference Center for Mycoses and Antifungals also reported that among the four Ascomycetes isolates resistant to all five classes of antifungal drugs, three were included or related to the *Y. lipolytica* clade, including one *Y. lipolytica* reference strain (21).

Our findings are in line with those of the literature, where *Y. lipolytica* typically exhibits high MICs for flucytosine. While there have been instances of reduced sensitivity to

**TABLE 3** *In vitro* antifungal susceptibility testing of 55 clinical isolates of *Yarrowia lipolytica* as determined by different methods[a]

| | | Anidulafungin | Micafungin | Caspofungin | Flucytosine | Posaconazole | Voriconazole | Itraconazole | Fluconazole | Amphotericin B |
|---|---|---|---|---|---|---|---|---|---|---|
| BMD | Range | 0.06–0.5 | 0.03–1 | 0.03–0.5 | 0.25–64 | 0.12–16 | 0.03–4 | 0.06–16 | 0.5–64 | 0.5–2 |
| | $MIC_{50}$ | 0.25 | 0.50 | 0.25 | 4 | 1 | 0.125 | 0.25 | 4 | 1 |
| | $MIC_{90}$ | 0.50 | 0.50 | 0.50 | 64 | 8 | 4 | 8 | 64 | 1 |
| | GM | 0.262 | 0.374 | 0.231 | 4.712 | 1.285 | 0.298 | 0.546 | 5.413 | 1.039 |
| SYO | Range | 0.03–0.5 | 0.12–1 | 0.06–1 | 0.25–128 | 0.12–16 | 0.03–8 | 0.03–32 | 2–128 | 0.5–2 |
| | $MIC_{50}$ | 0.125 | 0.5 | 0.25 | 8 | 0.5 | 0.125 | 0.25 | 4 | 1 |
| | $MIC_{90}$ | 0.25 | 0.5 | 0.5 | 64 | >8 | 4 | >16 | 128 | 2 |
| | GM | 0.123 | 0.398 | 0.279 | 10.963 | 1.076 | 0.209 | 0.719 | 8.849 | 1.052 |
| MTS | Range | 0.047–1.5 | 0.002–0.5 | 0.38–6 | 0.125–64 | 0.023–64 | 0.008–64 | 0.012–>32 | 0.75–>256 | 0.064–0.75 |
| | $MIC_{50}$ | 0.38 | 0.19 | 1.5 | >32 | 1 | 0.094 | 0.75 | 4 | 0.25 |
| | $MIC_{90}$ | 0.75 | 0.38 | 2 | >32 | 16 | 6 | >32 | >256 | 0.5 |
| | GM | 0.32 | 0.166 | 1.287 | 19.506 | 1.551 | 0.198 | 1.181 | 10.955 | 0.234 |
| ATB | Range | — | — | — | 4–>16 | — | 0.06–8 | 0.12–>4 | 2–128 | ≤0.5–1 |
| | $MIC_{50}$ | — | — | — | 16 | — | 0.125 | 0.25 | 4 | ≤0.5 |
| | $MIC_{90}$ | — | — | — | >16 | — | 4 | >4 | 64 | ≤0.5 |
| | GM | — | — | — | 10.293 | — | 0.293 | 0.579 | 6.791 | 0.519 |
| EA (%) | SYO | 98.18 | 98.18 | 100 | 61.82 | 100 | 100 | 85.45 | 100 | 100 |
| | MTS | 98.18 | 90.91 | 47.27 | 50.91 | 90.90 | 90.90 | 80 | 78.18 | 61.82 |
| | ATB | — | — | — | 83.64 | — | 98.18 | 98.18 | 100 | 98.18 |

[a]BMD: broth microdilution; SYO: Sensititre YeastOne; ATB: ATB FUNGUS 3; MTS: Liofilchem minimum inhibitory concentration (MIC) test strip; GM: geometric mean. To allow the calculation of geometric means, high off-scale MICs were raised to the next higher concentration. EA (%): percentage essential agreement (VS. BMD); "—" means no data available.

echinocandins, the isolates we tested were wild-type regarding echinocandin suscepti‐ bility. Amphotericin B generally shows low MICs against *Y. lipolytica*. However, a troubling increase in MICs for azole drugs has been observed, particularly in clinical isolates, suggesting an emerging resistance likely driven by antifungal drug exposure.

We suggest using L-ECOFFs to address the limitations of traditional ECOFFs that may not adequately reflect single-center laboratory conditions. In our analysis of the *ERG11* gene in 135 isolates, including global clinical isolates, we identified the *ERG11* A395T mutation in four strains from Chinese patients with bloodstream infections, all exhibiting azole MICs above the L-ECOFFs. This underscores the feasibility and effectiveness of L-ECOFFs.

In general, widespread use of azoles significantly increases exposure to azoles, leading to acquired resistance (22). In our study, all non-WT isolates were isolated from clinical samples, indicating potential exposure to antifungal drugs. Notably, the four *ERG11* A395T mutant strains had been treated with azole drugs, supporting the notion of acquired resistance, particularly to azoles, in *Y. lipolytica*. Although our data point toward this trend, further detailed studies are necessary for a conclusive understanding. This situation emphasizes the need for vigilant monitoring and prudent use of antifungal medications to avert the emergence and spread of resistant isolates.

Due to the absence of clinical breakpoints, categorical agreement could not be calculated. The EA rates of the clinical isolates are listed in Table 3. Our research shows that the results of the ATB and SYO have the highest EA with BMD, and almost all clinical isolates can be identified within 48 h. The SYO is an alternative for antifungal susceptibility testing and can test for nine antifungal drugs, including echinocandins (the ATB has not yet been able to conduct susceptibility testing for echinocandins), and exhibits excellent overall consistency. However, caution should be exercised when using the MTS to determine the antifungal susceptibility of *Y. lipolytica*. The results of the MTS need to be read after 48 h and are not easy to interpret, and the RPMI 1640 medium required for the test needs to be prepared in the laboratory and has a limited storage time. However, the MTS is recommended as a clone-screening method for studying heterogeneous drug resistance in experimental research.

Our study revealed that all three commercial methods were less accurate in determining susceptibility to flucytosine. Due to the propensity for rapid development of resistance, it is generally not recommended to use flucytosine as a monotherapy in clinical practice (23). For the necessary cases of susceptibility testing for flucytosine, we recommend the BMD method as the most reliable, ensuring optimal decision-making regarding the administration of flucytosine in clinical settings.

Although our study's sample size may seem small in comparison to those of other studies on more prevalent pathogens, however, we do not consider it as a limitation because *Y. lipolytica* being an emerging opportunistic yeast, the occurrence of clinical cases is naturally limited. Our research included all clinical isolates of *Y. lipolytica* from the CHIF-NET, representing the most extensive collection in China. Even a modest data set can be significant for rare fungi like *Y. lipolytica*, offering critical insights for clinical management (24, 25). Nonetheless, our study has certain limitations. It is essential to include clinical isolates from other global regions to ensure the applicability of ECOFFs. Furthermore, our study did not extensively explore the mechanisms underlying acquired drug resistance in *Y. lipolytica*, highlighting the necessity for further comprehensive research in the future.

This comprehensive study examined the antifungal susceptibility of *Y. lipolytica* across various Chinese locales over a decade, encompassing environmental, industrial, and food-related non-clinical isolates. This study is currently the largest to fill the gap in antifungal susceptibility profiles of *Y. lipolytica*. Notably, this is the first study to estab‐ lish and recommend L-ECOFFs for *Y. lipolytica* worldwide. Additionally, we evaluated the consistency between three prevalent commercial antifungal susceptibility testing methods and the BMD method. The SYO is the most recommended laboratory antifungal susceptibility testing method for *Y. lipolytica*, followed by the ATB, whereas the use

of the MTS method requires caution. We hope that this study will be of constructive significance for the clinical treatment of *Y. lipolytica* infections.

## MATERIALS AND METHODS

### Isolates

In this study, 55 clinical isolates were obtained from 22 hospitals in the China Invasive Fungal Resistance Surveillance Network (CHIF-NET, 96 hospitals in 29 provincial administrative regions) from 2009 to 2022, and 22 non-clinical isolates were also included. The sources of the isolates used in this study are shown in Fig. 2A. All isolates were accurately identified by matrix-assisted laser desorption ionization time-of-flight mass spectroscopy (MALDI-TOF) and sequencing of the nuclear ribosomal internal transcribed spacer (ITS) region, along with the large subunit of the 28S ribosomal DNA gene (D1/D2). *Candida parapsilosis* ATCC 22019, *Candida krusei* ATCC 6258, and *Candida albicans* ATCC 90028 were used as quality control strains in all tests.

### Antifungal susceptibility testing

The antifungal susceptibility testing process is illustrated in Fig. 2B. The initial concentration of the *Y. lipolytica* suspension was set at 0.5 or 2.0 McFarland using a turbidimeter (bioMérieux, Marcy-l'Étoile, France). The panels were inoculated and incubated at 35°C for 24–96 h.

### Broth microdilution method

The *in vitro* broth dilution susceptibility test was performed following the CLSI performance standard guidelines to detect the sensitivity of nine antifungal drugs (fluconazole, voriconazole, itraconazole, posaconazole, caspofungin, micafungin, anidulafungin, amphotericin B, and flucytosine) commonly used in a clinical setting (26). The MICs for the antifungal drugs were read at 50% as observed after 24 h of incubation (prominent decrease in turbidity or >50% inhibition of growth compared to the growth control), except for amphotericin B, which was read at 100% (27).

### Sensititre YeastOne YO10

The Sensititre YeastOne YO10 (SYO; Thermo Fisher Scientific, Waltham, MA, USA) panel was performed according to the manufacturer's instructions; 0.5 McFarland standard suspension (20 µL) was inoculated into 11 mL of YeastOne inoculum broth (Thermo Fisher Scientific, Waltham, MA, USA; Y3462). *Y. lipolytica* growth can be identified by a color change in the reaction well, typically from blue (negative) to red (positive). The MIC was determined by comparing the drug concentration in the first well with the obvious color change in the growth control well.

### ATB FUNGUS 3

The ATB FUNGUS 3 (ATB; bioMérieux, La Balme-les Grottes, France) strip was used to assess the antifungal susceptibility of *Y. lipolytica* to five antifungal drugs, namely, fluconazole, itraconazole, voriconazole, amphotericin B, and flucytosine, in semi-solid media under conditions similar to that of the BMD method. However, the ATB strip cannot detect susceptibility to echinocandins. The strip was filled with a suspension comprising ATB F2 medium and 20 µL of 2.0 McFarland suspension. The results were interpreted as follows: for amphotericin B, the MIC correspond to the lowest concentration enabling complete growth inhibition. For flucytosine, fluconazole, itraconazole, and voriconazole, the MIC was interpreted as the lowest concentration that results in distinct growth reduction or significant growth inhibition.

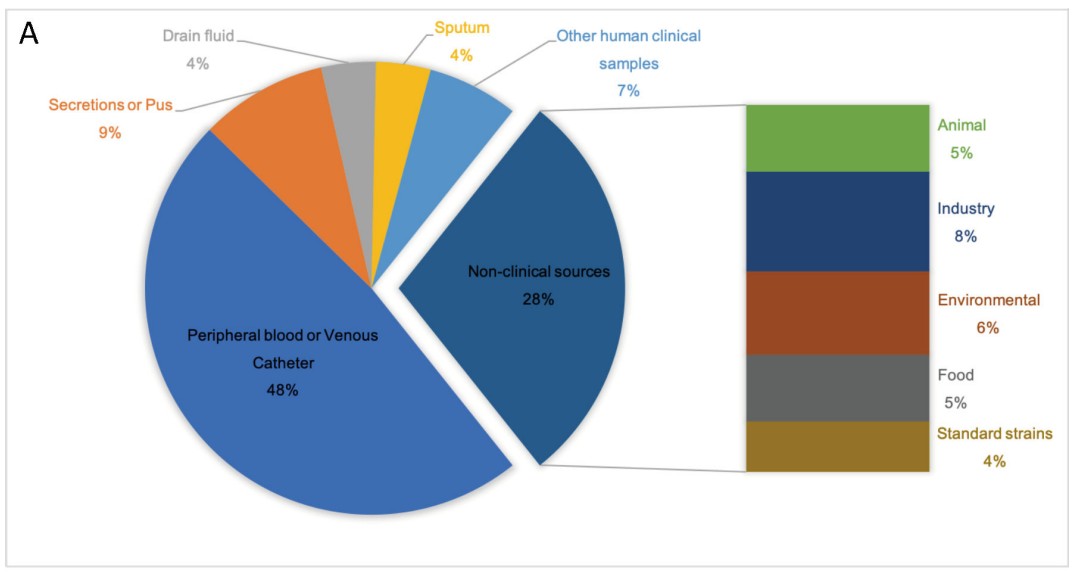

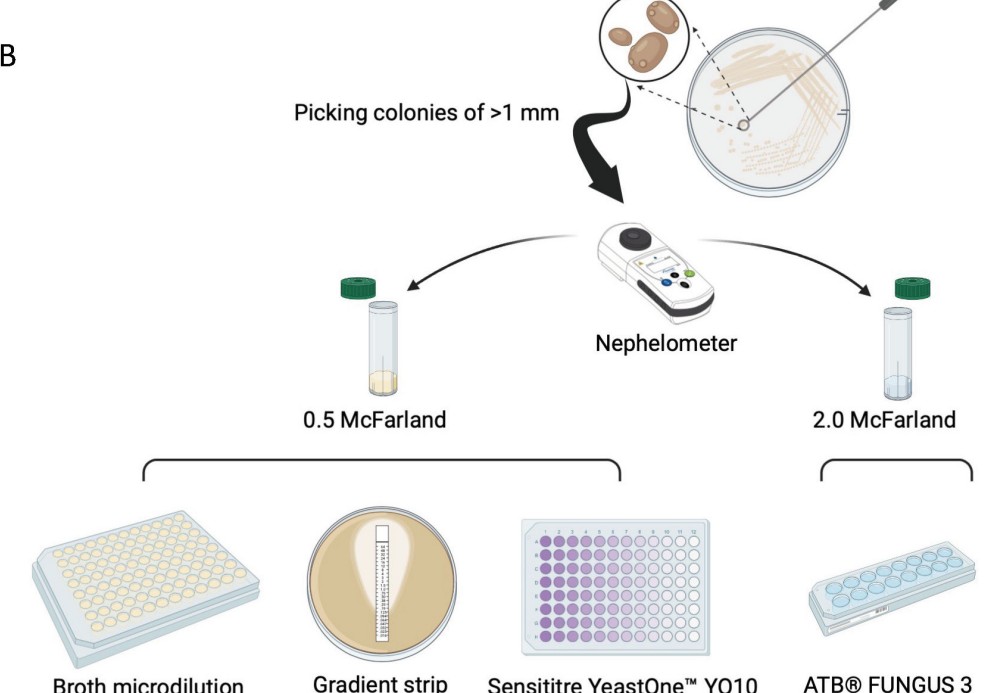

**FIG 2** Distribution of *Yarrowia lipolytica* sources analyzed in this research and protocol for antifungal susceptibility testing.

## Gradient strip

The MIC test strip (MTS; Liofilchem, Roseto degli Abruzzi, Italy) method was performed according to the manufacturer's instructions. The principle of this method is based on agar gradient diffusion. RPMI 1640 medium (with L-glutamine and without sodium bicarbonate) supplemented with glucose to a final concentration of 2% and 1.5% agar was used and buffered with 165 mM MOPS (pH 7.0). The MTS that contains a gradient of the antifungal drug concentration is placed on RPMI 1640 medium that has been inoculated with *Y. lipolytica*. After a certain period of incubation, the MIC can be directly read from the strip, which is the drug concentration at the junction of the inhibition zone and the strip.

## Mutation of *YALI0_B05126g* (*ERG11*)

The putative *ERG11* gene of *Y. lipolytica* is YALI0_B05126g (Gene ID:2906856), which was identified using the protein BLAST algorithm and yielded a protein of 523 amino acids that showed 59.21% similarity to its homolog in *Candida albicans* (GenBank accession number X13296.1). We designed amplification and sequencing primers for *YALI0_B05126g* using the National Center for Biotechnology Information (NCBI) Blast Primer tool (https://www.ncbi.nlm.nih.gov/tools/primer-blast/). The *Y. lipolytica ERG11* gene was amplified from 77 isolates using forward primer "TGATCATTCTCACGACGCTCA," and reverse primer "TCTAGTTACGCTCTCGCTTGC."

To thoroughly investigate the *ERG11* mutation spectrum in *Y. lipolytica*, we accessed next-generation sequencing genome data for 58 isolates, including all clinical isolates up to December 2021, from the NCBI Sequence Read Archive Database (https://www.ncbi.nlm.nih.gov/sra). The isolates included in the *YALI0_B05126g* analysis are shown in Table S3. The clean read sequences were aligned to the reference genome (ASM252v1) using the Bowtie 2 software (v2.3.5.1), and the alignment results were sorted using samtools (28, 29). The variation sites were annotated using the SnpEff software to determine gene information corresponding to the variant positions, synonymous/non-synonymous mutations, and their impact on amino acids (30).

## Establishment of L-ECOFFs and analysis

L-ECOFFs were determined for each drug and method using ECOFFinder XL 2010 v2.1 (https://clsi.org/meetings/susceptibility-testing-subcommittees/ecoffinder), based on 97.5% of the theoretical distribution (25, 31). One isolate was classified as wild type (WT) or non-WT when its MIC was less than or equal to the L-ECOFF and greater than the L-ECOFF, respectively. The essential agreement (EA) refers to the MIC obtained using the BMD method within two two-fold dilutions of the MIC values detected using the commercial method (32).

## ACKNOWLEDGMENTS

We are grateful to all participants in the National China Hospital Invasive Fungal Surveillance Network (CHIF-NET) program. We thank the teams of Wang Linqi and Bai Fengyan from the Institute of Microbiology, Chinese Academy of Sciences, for donating the environmental isolates of *Yarrowia lipolytica*.

## AUTHOR AFFILIATIONS

[1]Department of Clinical Laboratory, State Key Laboratory of Complex Severe and Rare Diseases, Peking Union Medical College Hospital, Peking Union Medical College, Chinese Academy of Medical Sciences, Beijing, China

[2]Beijing Key Laboratory for Mechanisms Research and Precision Diagnosis of Invasive Fungal Diseases, Beijing, China

[3]Graduate School, Peking Union Medical College, Chinese Academy of Medical Sciences, Beijing, China

[4]Department of Clinical Laboratory, Yongzhou Central Hospital, Yongzhou, China

[5]Department of Microbiology, the First Affiliated Hospital, Harbin Medical University, Harbin, China

[6]Department of Microbiology, the Fourth Affiliated Hospital, Harbin Medical University, Harbin, China

[7]Department of Clinical Laboratory, The Second Affiliated Hospital of Luohe Medical College, Luohe, China

## AUTHOR ORCIDs

Jinhan Yu  http://orcid.org/0000-0003-3807-2088
Yingchun Xu  http://orcid.org/0000-0002-7126-9459

Ying Zhao ⓘ http://orcid.org/0000-0002-7093-1121

## FUNDING

| Funder | Grant(s) | Author(s) |
|---|---|---|
| Chinese Academy of Medical Sciences Innovation Fund for Medical Sciences | 2021-I2M-1-038 | Yingchun Xu |
| National Natural Science Foundation of China (NSFC) | 82202592 | Ying Zhao |
| National high level hospital clinical research funding | 2022-PUMCH-C-052 | Ying Zhao |
| 北京市科学技术委员会 \| Natural Science Foundation of Beijing Municipality (Beijing Natural Science Foundation) | 7222125 | Ying Zhao |

## AUTHOR CONTRIBUTIONS

Jinhan Yu, Conceptualization, Data curation, Formal analysis, Investigation, Methodology, Writing – original draft, Writing – review and editing | Xueqing Liu, Data curation, Investigation, Methodology | Dawen Guo, Formal analysis, Investigation, Methodology, Resources | Wenhang Yang, Data curation, Formal analysis, Methodology | Xinfei Chen, Methodology, Resources | Guiling Zou, Methodology, Resources | Tong Wang, Methodology | Shichao Pang, Investigation, Methodology | Ge Zhang, Investigation | Jingjing Dong, Methodology | Yingchun Xu, Funding acquisition, Methodology, Resources, Supervision, Visualization, Writing – review and editing | Ying Zhao, Conceptualization, Data curation, Funding acquisition, Investigation, Methodology, Project administration, Visualization, Writing – original draft, Writing – review and editing

## DATA AVAILABILITY

The raw data supporting the conclusions of this article will be made available by the authors, without undue reservation.

## ETHICS APPROVAL

Ethical approval was not required as the study was conducted in a manner that allowed subjects to remain anonymous.

## ADDITIONAL FILES

The following material is available online.

### Supplemental Material

**Tables S1 to S3 (Spectrum03203-23-s0001.docx).** Supplemental tables.

### Open Peer Review

**PEER REVIEW HISTORY (review-history.pdf).** An accounting of the reviewer comments and feedback.

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
