## [Reviewer comments · Microbiology Spectrum]

Microbiology Spectrum

Antifungal susceptibility profile and local epidemiological cut-off values of *Yarrowia (Candida) lipolytica*: an emergent and rare opportunistic yeast

Jin-Han Yu, Xueqing Liu, Dawen Guo, Wen-Hang Yang, Xin-Fei Chen, Gui-Ling Zou, Tong Wang, Shichao Pang, Ge Zhang, Jingjing Dong, Ying-Chun Xu, and Ying Zhao

Corresponding Author(s): Ying Zhao, Peking Union Medical College Hospital

Review Timeline:

Submission Date:	August 28, 2023
Editorial Decision:	October 26, 2023
Revision Received:	November 10, 2023
Editorial Decision:	November 13, 2023
Revision Received:	November 14, 2023
Accepted:	November 15, 2023

Editor: Alexandre Alanio

Reviewer(s): The reviewers have opted to remain anonymous.

Transaction Report:

DOI: <https://doi.org/10.1128/spectrum.03203-23>

Re: Spectrum03203-23 (Antifungal susceptibility profile and local epidemiological cut-off values of *Yarrowia* (*Candida*) *lipolytica*: an emergent and rare opportunistic yeast)

Dear Prof. Ying Zhao:

Thank you for the privilege of reviewing your work. Below you will find my comments, instructions from the Spectrum editorial office, and the reviewer comments.

Revision Guidelines

Sincerely,
Alexandre Alanio
Editor
Microbiology Spectrum

Reviewer #1 (Comments for the Author):

The present manuscript describes the establishment of local ECOFFs for *Y. lipolytica* and compares three different commercial kits with the standard method according to CLSI. 74 isolates (both clinical and non clinical) were included in the study. In addition, the authors looked for mutation of ERG11.

The paper is interesting and data for *Y. lipolytica* is scarce. Therefore, more data is definitely important. The paper is well written, however, there are some inconsistencies which should be revised.

1. the number of isolates is very small. It is doubtful, if this number is sufficient for establishing sound ECOFFS. For the establishment of ECOFFS it is crucial to ensure that methodological variation (e.g. the variation created by using material from different manufacturers, incubation times varying between 16 and 20 hours, several staff members, thermostats), interlaboratory variation, and whatever biological variation there may be between isolates are incorporated into the MIC distribution and the ECOFF. It seems that these criteria are not completely fulfilled. The authors should therefore explain in a more detailed way how the results were obtained.

2. the comparison and evaluation of commercial kits is very important and valuable for diagnostic work up in routine laboratories. However, the rate of concordance and discordance should be stated explicitly. It is unclear how accuracy was calculated, more information should be given.

3. Specific comments:

Material and Methods:

Page 6, lines 123 +130, page 7, line 136: it would be easier to read if the exact name of the test would be stated and not the abbreviation

page 6, line 131: please insert City and Country of the company's name

Results:

page 8, lines 166-167: please explain why these strains did not yield any result. Was there no growth? What about the positive controls when testing these isolates? What could be the reason that you could not achieve any result?

Page 10, line 194: only caspofungin is mentioned, why are micafungin and anidulafungin missing?

Table 2 and page 9, lines 178 - 183: it is unclear if the results are from your work or if data from other isolates are included (135 isolates are quoted, but you tested 74 isolates). The information given is for readers without sound knowledge of ERG11 mutations not easy to understand. More explanations would therefore be needed.

Reviewer #2 (Comments for the Author):

Authors present an important dataset of antifungal susceptibility profiles for *Y. lipolytica* clinical and environmental isolates. This species is a rare opportunistic pathogen and antifungal profiles associated with some molecular data is helpful for understand potential resistance of those rare species. I don't understand why authors indicate that number of isolates is not sufficient. In my opinion, some sentences need clarification. Additional technical details are also essential in the materials and methods section.

Other remarks:

Line 26: *lipolytica* and not "lipolytic"

Authors indicate that the antifungal susceptibility profile of the *Y. lipolytica* is unknown but there are some studies that already give MIC distribution for isolates of these species and they mentioned some like ref 25 and 26. They could also add one reference with 34 isolates of *Y. lipolytica* for which antifungal MIC were determined (doi: 10.1111/myc.13095)

Line 40 "ATB could not detect echinocandins" do you mean that ATB is not good to determine susceptibility to echinocandins or that ATB is not able to distinguish between wild-type and non wild-type isolates?

Line 43: authors say that their study give information about epidemiology of *Y. lipolytica* but there is no clinical data presented in this study so I am not sure we could say that it's increase the knowledge concerning the epidemiology of this species.

Line 83 could you explain what do you mean by azoles are important for the treatment ? azoles are recommended, frequently used?

Line 87: not known is maybe more appropriate that unclear

Line 93: maybe also important to remind that commercialized methods are expensive

Line 102: could you give more details about the species identification? Were the isolates identified by MALDI ToF or sequencing or other techniques?

Line 111: Many isolates of *Y. lipolytica* don't grow well at 35{degree sign}C, did you test to determine antifungal susceptibility profile at 30{degree sign}C?

Line 114 for the strains with differing results did you redo the test or did you check the purity of the strains, is it possible that some samples contained mixture of isolates explaining the discordance?

Line 149 could you give more details concerning the method of extraction and sequencing because the paragraph is not clear for me, Did you sequence the region corresponding to the YAL10_B05126g gene for all your isolates?

Line 167: did you perform the tests at 30{degree sign}C for the isolates which did not grow at 35{degree sign}C?

Line 176 echinomycin is echinocandin?

Line 179 replace codon by nucleotide maybe

Line 181: "the two patients with isolated Y08...." Isolated corresponds to isolates?

Sometimes you indicate strains and in some other sentences isolates, could you harmonize the term

Line 192: could you clarify, do you mean ATB method could not detect echinocandin resistant ?

Line 206: some references could be interesting about the treatment of infection due to *Y. lipolytica*

In the references 25 the 5flucytosine is not tested and in the reference 26 the value of 5FC MIC are high so I don't understand why do you indicate that isolates have high MIC except for 5flucytosine.

Line 218: could you conclude that AMPHOB and voriconazole are the most effective treatment?

Lines 220-222 could you clarify the sentence

Line 224 did you sequence the entire ERG11 gene or only the coding region? Maybe sequenced is more appropriate than analysed.

Line 227 : Y132F is a mutation in the protein Erg11

Line 232 could you develop the last sentence

Line 246 the use of 5flucytpsine with other antifungal agents is frequent for many fungal species and not only *Y. lipolytica*

Line 249: Authors indicate that they determine antifungal profiles for 74 isolates which is a huge quantity for a rare opporutistic pathogen, why do they say that they have a restricted number of isolates?

November 10, 2023

Dear Reviewer,

Thank you for your thorough review, insightful comments, and constructive suggestions, all of which have considerably enhanced the presentation of our manuscript. We have meticulously revised the manuscript in line with your remarks.

Below, we provide a summary of our responses to each point raised. We trust that our revisions have adequately addressed all the issues raised, and we are hopeful that our manuscript is now suitable for publication.

Response to Reviewer 1

Comment 1:

the number of isolates is very small. It is doubtful, if this number is sufficient for establishing sound ECOFFS. For the establishment of ECOFFS it is crucial to ensure that methodological variation (e.g. the variation created by using material from different manufacturers, incubation times varying between 16 and 20 hours, several staff members, thermostats), interlaboratory variation, and whatever biological variation there may be between isolates are incorporated into the MIC distribution and the ECOFF. It seems that these criteria are not completely fulfilled. The authors should therefore explain in a more detailed way how the results were obtained.

Response : Thank you for bringing attention to the concerns about sample size and methodological variation in our study. We acknowledge these concerns and, after a comprehensive review of the standards for testing the susceptibility of rare fungal pathogens and establishing Epidemiological Cutoff Values (ECOFFs), we maintain that our findings are clinically significant. Despite the smaller sample size, which is a common challenge when dealing with rare fungi such as those with an isolation rate below 1%, the established ECOFFs remain pertinent, as detailed in the corresponding section of our article.

Moreover, it is important to highlight that our study encompasses all strains from China's most extensive fungal disease surveillance network, CHIF-NET, which spans the entire country. This wide coverage lends considerable reliability to our results concerning the drug susceptibility of *Yarrowia lipolytica* in China.

Addressing the issue of methodological variation, the rarity of *Yarrowia lipolytica* means not every laboratory has the necessary strains for testing, which presents a challenge for multi-laboratory studies. Nevertheless, we mitigated this by conducting three separate replicate experiments, carried out by different personnel, using freshly prepared media for each, and with results assessed blind to the experiment. While conducted in a single central laboratory, we used the term "Local ECOFF (L-ECOFF)" to align our terminology with similar studies, ensuring consistency and clarity in our findings.

Comment 2:

the comparison and evaluation of commercial kits is very important and valuable for diagnostic work up in routine laboratories. However, the rate of concordance and discordance should be stated explicitly. It is unclear how accuracy was calculated; more information should be given.

Response : Thank you for your astute suggestion, which we had initially overlooked. We have replaced any inappropriate terminology in the article, such as "accuracy," with more precise terms such as "essential agreement" or "concordance" to better reflect the intended meaning.

Comment 3:

Page 6, lines 123 +130, page 7, line 136: it would be easier to read if the exact name of the test would be stated and not the abbreviation

page 6, line 131: please insert City and Country of the company's name

Response : Thank you for your recommendation. I have amended the manuscript by spelling out the previously unclear abbreviations on Page 6, lines 123 and 130, and on Page 7, line 136, to their full terms for clarity. Additionally, the omitted city and country names at line 131 have been duly inserted.

Comment 4:

page 8, lines 166-167: please explain why these strains did not yield any result. Was there no growth? What about the positive controls when testing these isolates? What could be the reason that you could not

achieve any result?

Page 10, line 194: only caspofungin is mentioned, why are micafungin and anidulafungin missing?

Response : Thank you for your suggestions.

The antifungal susceptibility tests for the strains sourced from diverse locations—flies (CGMCC 2.3222), East China garbage stations (CGMCC 2.1556), and CGMCC 2.1502 from Kyoto University's Faculty of Agriculture in Japan—did not produce results due to their poor growth at 35°C. This outcome, which also affected the growth of positive controls, is consistent with the environmental nature of *Yarrowia lipolytica*, known for its limited tolerance to elevated temperatures. Following your advice, which was echoed by another reviewer, we have now carried out additional drug sensitivity assays at the more suitable temperature of 30°C for these three strains.

The results can be seen in **Supplementary Table 1**.

Supplementary Table 1. Antifungal susceptibility testing (performed at 30°C) of strains CGMCC 2.3222, CGMCC 2.1556, and CGMCC 2.1502.

	Anidulafungin	Micafungin	Caspofungin	5-flucytosine	Posaconazole	Voriconazole	Itraconazole	Fluconazole	Amphotericin B
CGMCC 2.3222	0.06	0.12	0.12	>64	1	0.12	0.5	16	0.5
CGMCC 2.1502	0.12	0.12	0.25	>64	0.5	0.5	0.25	32	0.5
CGMCC 2.1556	0.06	0.12	0.25	2	0.25	0.06	0.25	4	0.5

Note: Sensititre YeastOne™ YO10 (Thermo Fisher Scientific, Waltham, MA, USA) was used for the antifungal susceptibility testing.

The strain CGMCC 2.1502 is non-wild type for Voriconazole and Fluconazole, and CGMCC 2.3222 is also non-wild type for Fluconazole. This may also suggest that *Yarrowia lipolytica* naturally has insensitivity to some azole drugs.

In line 194 on Page 10, which we had previously overlooked, we have now added descriptions for the other two drugs, micafungin and anidulafungin.

Comment 5:

Table 2 and page 9, lines 178 - 183: it is unclear if the results are from your work or if data from other

isolates are included (135 isolates are quoted, but you tested 74 isolates). The information given is for readers without sound knowledge of ERG11 mutations not easy to understand. More explanations would therefore be needed.

Response : Thank you for your valuable feedback.

The fungal cytochrome P450 lanosterol 14 α -demethylase, encoded by the ERG11 gene, plays a crucial role in synthesizing ergosterol, unique to fungi, and serves as the target for azole drugs. Notably, the *ERG11* mutation (A395T) has been identified exclusively in the fluconazole-resistant *Y. lipolytica* strain CBS 18115. Its presence in other strains remains to be determined.

The presumed *ERG11* gene in *Y. lipolytica*, YALI0_B05126g (Gene ID:2906856), was pinpointed using the protein BLAST algorithm. This gene encodes a 523 amino acid protein sharing 59.21% similarity with the *Candida albicans* homolog (GenBank accession number X13296.1).

For gene amplification and sequencing, we crafted primers for YALI0_B05126g with the NCBI Blast Primer tool. The ERG11 gene was amplified from 77 strains using the forward primer "TGATCATTCTCACGACGCTCA," and the reverse primer "TCTAGTTACGCTCTCGCTTGC."

To thoroughly investigate the *ERG11* mutation spectrum in *Y. lipolytica*, we accessed next-generation sequencing genome data for 58 strains, including all clinical isolates up to December 2021, from the NCBI Sequence Read Archive Database. The strains analyzed for YALI0_B05126g are listed in **Supplementary Table S1**. We aligned the clean read sequences with the reference genome (ASM252v1) using bowtie 2 software (v2.3.5.1), sorted the alignments with samtools, and annotated variation sites with SnpEff software to identify gene variants, categorize mutations as synonymous or non-synonymous, and assess their impact on the amino acids.

Response to Reviewer 2

Comment 1:

I don't understand why authors indicate that number of isolates is not sufficient.

Line 249: Authors indicate that they determine antifungal profiles for 74 isolates which is a huge

quantity for a rare opportunistic pathogen, why do they say that they have a restricted number of isolates?

Response : We apologize for any lack of clarity in our previous description.

When referring to a small sample size, we meant it in relation to other studies that establish Epidemiological Cutoff Values (ECOFFs) for more commonly encountered clinical pathogens. Nevertheless, we do not view this as a limitation in the context of our research. Given that *Y. lipolytica* is a rare and emerging opportunistic yeast, a low prevalence is inherent. Our study is comprehensive in that it includes clinical isolates of *Y. lipolytica* gathered from the National China Hospital Invasive Fungal Surveillance Network (CHIF-NET) over the last decade, which is the largest network of its kind in China. Moreover, in cases involving rare fungi like *Y. lipolytica*, even a seemingly small dataset can significantly inform clinical treatments, as supported by literature. In essence, for rare fungi, the sample size we have is considered adequate.

Comment 2:

Additional technical details are also essential in the materials and methods section.

Line 26: lipolytica and not "lipolytic"

Line 87: not known is maybe more appropriate than unclear

Line 93: maybe also important to remind that commercialized methods are expensive

Line 176 echinomycin is echinocandin?

Line 179 replace codon by nucleotide maybe

Line 227 Y132F is a mutation in the protein Erg11

Sometimes you indicate strains and in some other sentences isolates, could you harmonize the term

Response: Thank you for your insightful writing suggestions; they have been incredibly beneficial.

We have thoroughly reviewed and revised the manuscript, clarifying any ambiguous descriptions and correcting the inaccuracies you pointed out. Additionally, we have enriched the methodology section with a comprehensive account of the antifungal susceptibility tests and refined the approach for analyzing the *ERG11* mutations.

Our distinction between "strains" and "isolates" is well-founded and aligns with the terminology used

within the scientific literature. An isolate is indeed something that has been obtained from a specific source, such as an animal or a patient. Subsequent investigation allows us to classify these isolates, assigning a name and identifying them as a particular strain. Hence, the use of both terms in our manuscript is appropriate and accurately reflects the process of identification and classification within your research.

Comment 3:

Authors indicate that the antifungal susceptibility profile of the *Y. lipolytica* is unknown but there are some studies that already give MIC distribution for isolates of these species and they mentioned some like ref 25 and 26. They could also add one reference with 34 isolates of *Y. lipolytica* for which antifungal MIC were determined (doi: 10.1111/myc.13095)

Response: Thank you for your insightful suggestions. In our manuscript, the article 'Mycoses. 2020;63(7):737-745. doi:10.1111/myc.13095' is indeed acknowledged as the ninth citation. We recognize that the literature contains reports on the antifungal susceptibility of *Yarrowia lipolytica*; these reports tend to be case studies involving a limited number of strains. Critically, there appears to be an absence of studies employing the CLSI standard broth microdilution method. This absence casts doubts on the reliability of the reported drug sensitivity findings. Taking your advice into account, we agree that the term 'unclear' is more fitting than 'unknown' in our manuscript. This alteration more accurately conveys the present knowledge and underscores the necessity for standardized methodological practices in forthcoming studies.

Comment 4:

Line 246 the use of 5flucytpsine with other antifungal agents is frequent for many fungal species and not only *Y. lipolytica*.

Line 43: authors say that their study give information about epidemiology of *Y. lipolytica* but there is no clinical data presented in this study so I am not sure we could say that it's increase the knowledge concerning the epidemiology of this species.

Response: Thank you for your valuable writing advice; it has been greatly beneficial.

We have revised reads: Due to the propensity for rapid development of resistance, it is generally not recommended to use 5-flucytosine as a monotherapy in clinical practice.

In our manuscript, the use of 'epidemiology' was intended to denote the variability and distribution of drug sensitivity among *Yarrowia lipolytica* strains. We accept that this term may not be ideally suited, and we will adjust the text in our manuscript to better reflect our intended meaning.

Comment 5:

Line 40 "ATB could not detect echinocandins" do you mean that ATB is not good to determine susceptibility to echinocandins or that ATB is not able to distinguish between wild-type and non wild-type isolates?

Line 192: could you clarify, do you mean ATB method could not detect echinocandin resistant?

Response : We apologize for any lack of clarity in our initial description. ATB is suitable for *Candida* and *Cryptococcus* species and can detect susceptibility to five antifungal drugs: 5-flucytosine, amphotericin B, fluconazole, itraconazole, and voriconazole. However, it cannot detect susceptibility to echinocandins. Additionally, it cannot perform drug susceptibility testing for *Aspergillus* species. We will provide a detailed description in the manuscript to avoid any ambiguity.

Comment 6:

Line 83 could you explain what do you mean by azoles are important for the treatment? azoles are recommended, frequently used?

Response : Thank you for your recommendations.

Azole antifungals, including fluconazole, itraconazole, voriconazole, and posaconazole, function by disrupting ergosterol synthesis within fungal cell membranes, leading to cellular death. Their efficacy spans a wide spectrum of fungal species, establishing them as a frequent choice for treating various fungal infections.

Reports indicate that azoles are effective in treating *Yarrowia lipolytica* infections, with clinical

improvements observed upon administration. Consequently, their use is significant in managing invasive *Y. lipolytica* infections.

Comment 7:

Line 102: could you give more details about the species identification? Were the isolates identified by MALDI ToF or sequencing or other techniques?

Response : Thank you for your suggestions.

For precise identification, we utilized matrix-assisted laser desorption ionization time-of-flight mass spectrometry (MALDI-TOF MS) and sequenced the internal transcribed spacer (ITS) region of the nuclear ribosomal DNA, as well as the large subunit of the 28S ribosomal DNA gene (D1/D2).

Comment 8:

Line 111: Many isolates of *Y. lipolytica* don't grow well at 35°C, did you test to determine antifungal susceptibility profile at 30°C?

Line 167: did you perform the test at 30°C for the isolates which did not grow at 35°C?

Response: Thank you for your input.

The antifungal susceptibility tests for strains isolated from flies (CGMCC 2.3222), garbage stations in East China (CGMCC 2.1556), and strain CGMCC 2.1502 from Kyoto University's Faculty of Agriculture, Japan, were inconclusive due to their poor growth at 35°C. Consequently, positive controls also exhibited negligible growth at this temperature during the tests.

This outcome is not unexpected as *Yarrowia lipolytica* is an environmental fungus, with some strains unable to endure elevated temperatures.

In response to your suggestion, we have conducted the drug susceptibility testing for these strains at a lower temperature of 30°C.

Supplementary Table 1. Antifungal susceptibility testing (performed at 30°C) of strains CGMCC 2.3222, CGMCC 2.1556, and CGMCC 2.1502.

Anidulafungin	Micafungin	Caspofungin	5- flucytosine	Posaconazole	Voriconazole	Itraconazole	Fluconazole	Amphotericin B
---------------	------------	-------------	-------------------	--------------	--------------	--------------	-------------	-------------------

CGMCC 2.3222	0.06	0.12	0.12	>64	1	0.12	0.5	16	0.5
CGMCC 2.1502	0.12	0.12	0.25	>64	0.5	0.5	0.25	32	0.5
CGMCC 2.1556	0.06	0.12	0.25	2	0.25	0.06	0.25	4	0.5

Note: Sensititre YeastOne™ YO10 (Thermo Scientific, USA) was used for the antifungal susceptibility testing.

The strain CGMCC 2.1502 is non-wild type for Voriconazole and Fluconazole, and CGMCC 2.3222 is also non-wild type for Fluconazole. This may also suggest that *Yarrowia lipolytica* naturally has insensitivity to some azole drugs.

Comment 9:

Line 114 for the strains with differing results did you redo the test or did you check the purity of the strains, is it possible that some samples contained mixture of isolates explaining the discordance?

Response: Thank you for your guidance.

In our research, we sourced all strains from individual clones, which were then propagated to confirm the culture's purity. We repeated the experiments multiple times to verify the consistency of our results and to reduce potential variability arising from disparate reagent batches, variations in technique among different researchers, and possible inaccuracies in recording or interpreting the drug sensitivity data.

Comment 10:

Line 149 could you give more details concerning the method of extraction and sequencing because the paragraph is not clear for me, did you sequence the region corresponding to the YALI0_B05126g gene for all your isolates?

Line 224 did you sequence the entire ERG11 gene or only the coding region? Maybe sequenced is more appropriate than analysed.

Response: Thank you for your recommendations.

In our research, we approached the analysis of the *ERG11* gene from two ways. First, we extracted

genomic DNA from the 77 strains carried out amplification sequencing to acquire the gene sequences. Secondly, for the 58 strains from which we could not retrieve physical samples, we employed bioinformatics methods to extract ERG11 gene sequences from genomic data available in databases, leveraging the high accuracy of modern bioinformatics tools. We then collectively analyzed these two data sets of ERG11 gene sequences.

Using the National Center for Biotechnology Information (NCBI) Blast Primer tool, we developed primers for amplification and sequencing of YALI0_B05126g. We amplified the *Y. lipolytica* ERG11 gene from 77 strains using the forward primer "TGATCATTCTCACGACGCTCA," and the reverse primer "TCTAGTTACGCTCTCGCTTGC."

To thoroughly investigate the mutation spectrum of the ERG11 gene in *Y. lipolytica*, we accessed next-generation sequencing genome data for 58 strains, which includes all clinical isolates as of December 2021, from the NCBI Sequence Read Archive Database. The strains analyzed for YALI0_B05126g are detailed in Supplementary Table S1. We aligned the sequences to the reference genome (ASM252v1) using bowtie 2 software, organized the alignment data with samtools, and annotated variation sites using SnpEff software, which helped identify gene variations, discern synonymous from non-synonymous mutations, and assess their impact on amino acids.

Comment 11:

Line 181: "the two patients with isolated Y08...." Isolated corresponds to isolates?

Response: We apologize for any ambiguity in our description. The patient harboring the Y04 strain underwent an 11-day itraconazole treatment regimen. Additionally, the patient from whom the Y08 strain was isolated, as well as the patients associated with the Y26 and Y27 strains, had been treated with intravenous fluconazole.

Comment 12:

Line 206: some references could be interesting about the treatment of infection due to *Y. lipolytica*
In the references 25 the 5-flucytosine is not tested and in the reference 26 the value of 5FC MIC are

high so I don't understand why do you indicate that isolates have high MIC except for 5flucytosine.

Lines 220-222 could you clarify the sentence

Response: Thank you for your advice.

Yarrowia lipolytica infections are uncommon, leading to a scarcity of literature on their treatment, which is largely composed of case reports. Azole medications are commonly adopted as the initial treatment strategy clinically. Should azole therapy prove ineffective, treatment often transitions to echinocandins.

Pertaining to reference 25, an oversight in citation software management resulted in a misordering of references. We have thoroughly reexamined and verified each citation within our manuscript and have submitted a rectified version.

Furthermore, in the discussion section, specifically lines 220-222, we have incorporated the revised text.

Comment 13:

Line 218: could you conclude that AMB and voriconazole are the most effective treatment?

Line 232 could you develop the last sentence

Response: Thank you for your insights.

Y. lipolytica generally remains sensitive to Amphotericin B and Voriconazole. However, I contend that Amphotericin B is linked with more pronounced side effects. For *Y. lipolytica*, SYO can be utilized for sensitivity assessments, and azole antifungals are advisable for monotherapy due to their efficacy and tolerability.

Line 232: In our study, all non-WT strains were isolated from clinical samples, indicating potential exposure to antifungal drugs. Additionally, the four ERG11 mutant strains had a history of azole drugs treatment. Based on these findings, it is reasonable to speculate that *Y. lipolytica* has acquired antifungal resistance, specifically cross-resistance to azoles. Further research and analysis would be needed to definitively confirm this hypothesis. This highlights the importance of monitoring and managing the use of antifungal drugs to prevent the emergence and spread of resistant strains.

北京协和医院
PEKING UNION MEDICAL COLLEGE HOSPITAL

Peking Union Medical College Hospital
No.1 Shuaifuyuan Wangfujing Dongcheng District, Beijing, China 100730

Re: Spectrum03203-23R1 (Antifungal susceptibility profile and local epidemiological cut-off values of *Yarrowia (Candida) lipolytica*: an emergent and rare opportunistic yeast)

Dear Prof. Ying Zhao:

Thank you for the privilege of reviewing your work. Below you will find my comments, instructions from the Spectrum editorial office, and the reviewer comments.

I acknowledge the work done to reply to reviewers' comments.

The authors should restrict the use of strain when they are sure that a single colony has been re-isolated from the clinical isolate as a strain should have a defined genotype. A clinical isolate may contain more than one strain (several genotypes coexisting in the clinical sample). If the wording is appropriate considering this definition, please make it clearer in the methods.

Revision Guidelines

Sincerely,
Alexandre Alanio
Editor
Microbiology Spectrum

November 14, 2023

Dear Reviewer,

Thank you for your thorough review, insightful comments, and constructive suggestions, all of which have considerably enhanced the presentation of our manuscript. We have meticulously revised the manuscript in line with your remarks.

We trust that our revision has adequately addressed all the issues raised, and we are hopeful that our manuscript is now suitable for publication.

Response to Reviewer

Comment:

The authors should restrict the used of strain when they are sure that a single colony have been re-isolated from the clinical isolate as a strain should have a define genotype. A clinical isolate may contain more than one strain (several genotypes coexisting in the clinical sample). If the wording is appropriate considering this definition, please make it clearer in the methods.

Response: Thank you for your insightful writing suggestion; It have been incredibly beneficial.

We have thoroughly reviewed and correcting the inaccuracies you pointed out. The word "strain" is used when we refer to the define genotype. I hope we have understood your suggestion correctly.

Re: Spectrum03203-23R2 (Antifungal susceptibility profile and local epidemiological cut-off values of *Yarrowia (Candida) lipolytica*: an emergent and rare opportunistic yeast)

Dear Prof. Ying Zhao:

Your manuscript has been accepted, and I am forwarding it to the ASM production staff for publication. Your paper will first be checked to make sure all elements meet the technical requirements. ASM staff will contact you if anything needs to be revised before copyediting and production can begin. Otherwise, you will be notified when your proofs are ready to be viewed.

Sincerely,
Alexandre Alanio
Editor
Microbiology Spectrum